# Gain-of-function genetic screens in human cells identify SLC transporters overcoming environmental nutrient restrictions

Manuele Rebsamen[1,2,]*, Enrico Girardi[1,]*, Vitaly Sedlyarov[1], Stefania Scorzoni[1], Konstantinos Papakostas[1], Manuela Vollert[1], Justyna Konecka[1], Bettina Guertl[1], Kristaps Klavins[1], Tabea Wiedmer[1], Giulio Superti-Furga[1,3]

Solute carrier (SLC) transporters control fluxes of nutrients and metabolites across membranes and thereby represent a critical interface between the microenvironment and cellular and subcellular metabolism. Because of substantial functional overlap, the interplay and relative contributions of SLCs in response to environmental stresses remain poorly elucidated. To infer functional relationships between SLCs and metabolites, we developed a strategy to identify SLCs able to sustain cell viability and proliferation under growth-limiting concentrations of essential nutrients. One-by-one depletion of 13 amino acids required for cell proliferation enabled gain-of-function genetic screens using a SLC-focused CRISPR/Cas9–based transcriptional activation approach to uncover transporters relieving cells from growth-limiting metabolic bottlenecks. Among the transporters identified, we characterized the cationic amino acid transporter *SLC7A3* as a gene that, when up-regulated, overcame low availability of arginine and lysine by increasing their uptake, whereas SLC7A5 was able to sustain cellular fitness upon deprivation of several neutral amino acids. Moreover, we identified metabolic compensation mediated by the glutamate/aspartate transporters SLC1A2 and SLC1A3 under glutamine-limiting conditions. Overall, this gain-of-function approach using human cells uncovered functional transporter-nutrient relationships and revealed that transport activity up-regulation may be sufficient to overcome environmental metabolic restrictions.

## Introduction

A major cellular metabolic requirement is the ability to respond and adapt to the external environment to obtain sufficient amounts of the building blocks and sources of energy required for growth and proliferation. When the concentration of a critical nutrient decreases below a certain threshold because of lack of availability or increased consumption, transcriptional and metabolic programs are triggered to limit anabolic and promote catabolic processes via expression of metabolic enzymes or induction of autophagy and macropinocytosis ([1], [2], [3], [4]). Limiting nutrient availability in the extracellular microenvironment has been shown to deeply affect cell growth, activity, and proliferation in both normal and pathological contexts, including development ([5]), cell differentiation ([6], [7]), immune responses ([8]), and cancer microenvironment ([3], [9]). Amino acids in particular represent ~60% of the dry mass of a cell ([2]) and a major source of both biomass and energy for mammalian cells ([10]). Importantly, several amino acids, i.e., histidine, isoleucine, leucine, lysine, methionine, phenylalanine, threonine, tryptophan, and valine ("essential" amino acids), cannot be directly synthetized by cells and have therefore to be imported from the environment. This requires the action of transmembrane transporters located either at the plasma membrane or, in case of uptake via macropinocytosis or receptor-mediated endocytosis, in the endolysosomal compartment. Similarly, upon starvation, cells induce autophagy to restore nutrient and amino acid levels, and transporters in the autophagolysosome mediate their transfer to the cytoplasm ([2], [11]). Limiting concentrations of amino acids have been reported as metabolic liabilities of cancer cells, including glutamine ([12]) and cysteine ([13]) depletion in several solid cancers, asparagine in acute lymphoblastic leukemia ([14], [15]), and arginine in argininosuccinate synthetase–deficient malignancies ([15], [16]). Accordingly, reduction of the expression of amino acid transporters has been shown to affect cancer cell growth ([17], [18], [19]), and mouse cancer models showed a tumor-intrinsic dependency on glucose and amino acid transporters ([20], [21], [22], [23]). Despite these findings, it is not clear if changes in expression of individual plasma membrane transporters

[1]CeMM Research Center for Molecular Medicine of the Austrian Academy of Sciences, Vienna, Austria   [2]Department of Immunobiology, University of Lausanne, Epalinges, Switzerland   [3]Center for Physiology and Pharmacology, Medical University of Vienna, Vienna, Austria

Correspondence: gsuperti@cemm.oeaw.ac.at; manuele.rebsamen@unil.ch
Enrico Girardi's present address is Solgate GmbH, IST Park, Klosterneuburg, Austria.
Konstantinos Papakostas' present address is ViruSure GmbH, Tech Gate Vienna, Vienna, Austria.
*Manuele Rebsamen and Enrico Girardi contributed equally to this work.

alone would be sufficient to relieve cells from metabolic bottlenecks caused by limiting amount of nutrients.

Solute carriers represent the largest family of human transporters, counting more than 450 genes grouped in subfamilies based on sequence similarity (24, 25). The fact that a cell expresses at any given time a set of 150–250 solute carriers (26, 27), often with overlapping substrate specificities, results in a high level of functional complexity which hampers efforts to define the individual roles of each protein in nutrient uptake and cell growth. Moreover, cells are known to rearrange the repertoire of expressed transporters upon environmental challenges (28) and therapeutic treatments (29).

Forward genetics approaches have been particularly successful in assigning a gene to specific biological processes and pathways, both at the genome-wide level and with more focused approaches (30, 31, 32, 33). Most of the studies published so far focus on loss-of-function (lof) approaches as these tend to give binary, easily interpretable, outputs (i.e., wt versus ko phenotypes). However, the availability of CRISPR/Cas9–based approaches has recently expanded the range of readouts and phenotypes that can be explored with large, unbiased gain-of-function (gof) screens via up-regulation of genes from their endogenous loci (34, 35, 36). Importantly, gof approaches have the advantage of overcoming the issue of redundancy between genes and provide a ranking of the genes that, when overexpressed, confer a selective advantage in a given situation. This aspect is particularly attractive in the context of solute carriers, which have a high degree of substrate and functional redundancy among them (37).

Here, we present the development of a gof genetic screening approach that allowed us to systematically identify key solute carriers able to overcome nutrient limitation in intact human cells, where the transporters are within natural membranes, with their endogenous modifications and partners. By applying this approach to assess different transporter-based responses to limiting concentrations of 13 amino acids required in our cellular experimental system, we identified SLC7A3 (CAT3), SLC1A2/A3 (EEAT2/1), and SLC7A5 (LAT1) as the transporters able to rescue cells from low availability of arginine/lysine, glutamine, and histidine/tyrosine, respectively.

## Results

### Amino acid transporters are up-regulated upon single–amino acid starvation

To determine the cellular programs activated by cells upon nutrient starvation and the corresponding regulation of SLCs expression, we generated a transcriptional profile of HEK293T and HeLa cells upon removal of a specific nutrient. HEK293T cells were incubated for 16 h in defined media containing dialyzed serum (MWCO 10,000 D) and lacking a specific amino acid, vitamins, or glucose (Table S1) (37). As amino acid starvation impairs mTORC1 activity, we also treated cells with the mTOR inhibitor torin 1 (38). To evaluate general and cell type–specific effects, responses to single–amino acid depletion were also assessed in HeLa cells. Transcript abundance was determined

by 3′ mRNA sequencing, and enrichment was calculated by comparing the samples under nutrient-limiting conditions to the fully reconstituted media. We observed large transcriptional changes in most samples, with patterns consistent across the two cell lines (Figs 1A and B and S1A and B). One exception was observed upon deprivation of serine or glycine, which showed only marginal changes, likely as a result of the cell's ability to readily interconvert these two amino acids (39, 40). Interestingly, the absence of vitamins, as tested in HEK293T cells, did not result in large transcriptional changes in this experimental setup. Moreover, although we observed comparable changes in most of the other single–amino acid–depleted conditions, methionine depletion appeared to induce a distinct transcriptional response, similar to previous reports (41). To identify key transcription factors involved in the response to nutrient limitation, we calculated the enrichment of target genes in the tested samples based on the TRRUST dataset (42). We observed the ATF4 transcription factor program as the top enriched among most amino acid–depleted samples (Figs 1C and S1C), consistent with the role of this transcription factor downstream of the amino acid sensor GCN2 in activating the amino acid response (AAR) (28). We also observed enrichment for CEBPB targets, another transcription factor containing AAR elements, upon cysteine deprivation and consistent with previous reports describing CEBPB up-regulation in other cell lines (43). In line with AAR induction, we detected extensive changes in the mRNA abundance of amino acid transporters under conditions of limiting amino acid compared with fully reconstituted media (Figs 1D and S1D). More than 60 amino acid transporters have been described, comprising members of the SLC1, SLC7, SLC36, SLC38, and SLC43 subfamilies (44). Interestingly, a subset of transporters appeared to be highly up-regulated in most aa-limiting conditions, regardless of the amino acid removed from the media and consistent with the induction of AAR (Figs 1D and S1D). Within this set of SLCs, we observed several amino acid transporters, including members of the *SLC1* (*SLC1A4* and *SLC1A5*), *SLC7* (*SLC7A1, SLC7A3, SLC7A5, SLC7A11*, and the associated *SLC3A2* heavy chain), and *SLC38* families (*SLC38A1* and *SLC38A2*). Of note, several of these transporters are controlled by ATF4, including *SLC1A5* (*ASCT2*), *SLC7A1* (*CAT1*), and *SLC7A11* (*xCT*), and have been reported to play a role in maintaining amino acid homeostasis (18, 45). These results confirmed that a profound adaptation of transporter expression is a prominent phenomenon in the responses induced to overcome nutrient starvation and amino acid limitation in particular. Most importantly, this raises the question whether the observed changes in transporter expression are sufficient to overcome nutrient limitations and which of these SLCs may contribute most to restore cellular fitness in each of these conditions.

### Generation of a SLC-focused gof library and development of nutrient-dependent cellular fitness screens

To systematically interrogate the SLC superfamily and identify specific transporters involved in the response to nutrient limitation, we generated a gof lentiviral-based CRISPR/Cas9 library targeting 388 human SLCs (SLC-SAM library, Figs 2A and S2A and Table S2). The library was based on the synergistic activation mediator (SAM) approach (35), which has been shown to be one of the most effective CRISPR-based approaches developed for transcriptional activation (34). This approach relies on a nuclease-inactive Cas9

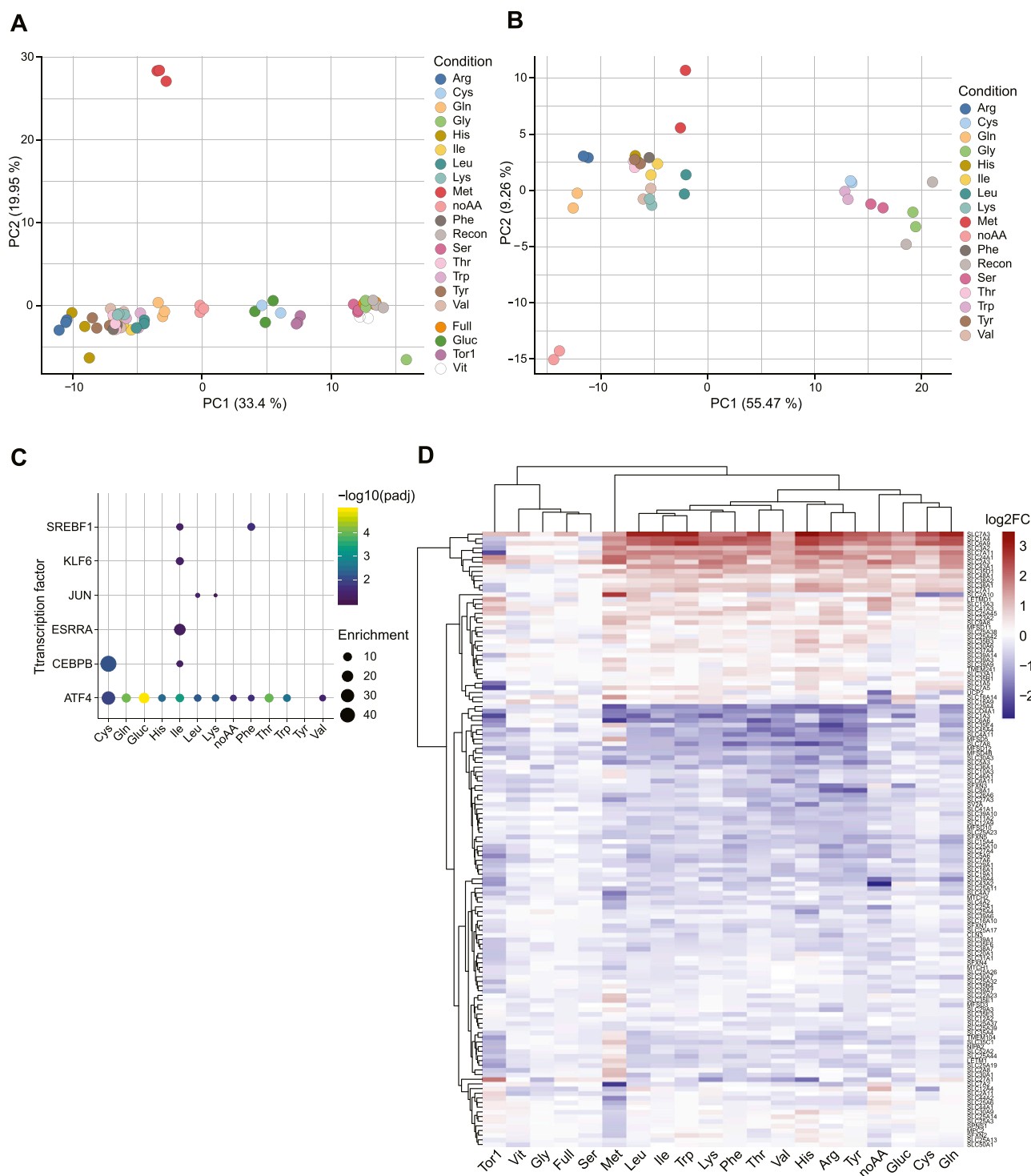

**Figure 1. Transcriptional responses upon single-nutrient starvation.**
**(A)** PCA plot of the transcriptomics profiles of HEK293T cells upon specific nutrient deprivation conditions or Torin-1 treatment for 16 h. Replicates for each condition are shown with the same color code. **(B)** PCA plot of the transcriptomics profiles of HeLa cells upon specific nutrient deprivation conditions for 16 h. Replicates for each condition are shown with the same color code. **(C)** Enrichment analysis for target genes of transcription factors in the HEK293T samples, as determined by TRRUST v2. **(D)** Heatmap showing differential expression of SLC transporters in the listed conditions compared with reconstituted media in HEK293T cells.

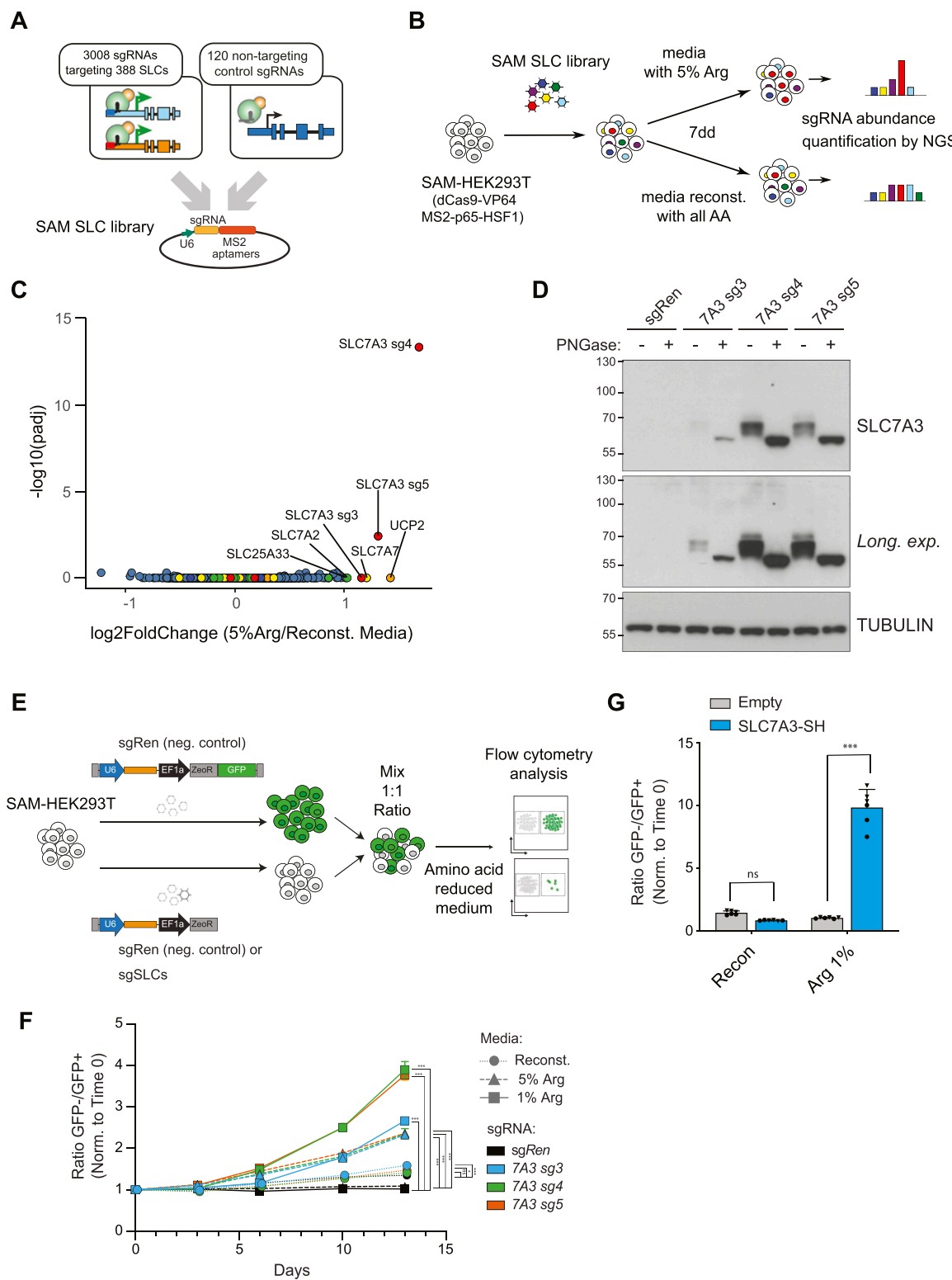

**Figure 2. A SLC-focused gain-of-function approach identifies SLC7A3 in arginine limiting conditions.**
**(A)** Schematic representations of the SLC synergistic activation mediator (SAM) library used in this study. **(B)** Experimental setup for the screening of SLCs able to overcome arginine limitations in the extracellular media. **(C)** Volcano plot showing enrichment of sgRNAs in arginine-limiting conditions against fully reconstituted media after 7 d of treatment. Log2 fold changes and adjusted $P$-values were calculated using DESeq and GSEA from two replicate experiments. **(D)** Western blot showing increased protein levels of SLC7A3 in SAM-HEK239T cells carrying sgRNA targeting *SLC7A3* compared with control sgRNA *Renilla*. Where indicated, cell lysates were treated with PNGase. **(E)** Schematic representation of the color competition assay (CCA) approach. **(F)** Time-dependent enrichment of SAM-HEK239T cells carrying sgRNAs targeting

fused to a transcriptional activator domain (VP64) and MS2-based recruitment of p65-HSF1 to induce single guide RNA (sgRNA)–driven overexpression of a given gene from its endogenous promoter. Up to six sgRNAs per SLC transcript were selected. Representation within the library was determined by next-generation sequencing (NGS, Fig S2B).

To perform genetic screens, we set out to identify conditions in which reduced levels of specific amino acids would confer a sufficient selective pressure. Thus, HEK293T were grown in amino acid–free media reconstituted with all amino acids at their original concentration with the exception of one particular amino acid. We chose to test the setup using arginine, which was supplemented at 0.02 mM, a concentration corresponding to 5% of the one present in normal, rich media (0.4 mM). Media was further supplemented with dialyzed serum. As expected, this resulted in reduced proliferation and cell death (Fig S2C), thereby conferring the selection pressure required for the planned gof screen. We therefore selected this limiting arginine condition to test whether the SLC-SAM library we generated allowed for the unbiased identification of transporters able to restore cell proliferation under nutrient starvation. We infected HEK293T stably expressing the SAM transcriptional activation machinery (SAM-HEK293T) with the SLC-SAM library and, after antibiotic selection, switched to media containing either 5% arginine or fully reconstituted. After 7 d in culture, the remaining cells were collected and the composition of the library determined by sequencing (Fig 2B and Table S3). Enrichment analysis of the samples grown in low-arginine versus the fully reconstituted conditions showed a strong and statistically significant enrichment for sgRNAs targeting the sodium-independent cationic amino acid transporter SLC7A3 (Fig 2C). This protein is able to transport arginine, as well as lysine and ornithine, with K$m$ in the micromolar range (46), therefore strongly supporting the ability of our approach to identify direct transporter/substrate relationships. Of note, several members of the SLC7 (CAT) subfamily of cationic amino acid transporters were indeed enriched in our transcriptomics analysis (Fig 1D).

### SLC7A3 up-regulation overcomes arginine limitation

To validate the previous results, we next generated cells individually expressing the three enriched sgRNAs targeting the SLC7A3 (sgSLC7A3) promoter or a control sgRNA encoding a Renilla sequence (sgRen). All SLC7A3 sgRNAs led to an up-regulation of SLC7A3 mRNA and protein levels (Figs 2D and S2D). To functionally assess whether increased levels of SLC7A3 confer resistance to reduced levels of arginine, we used color competition assay (CCA) (Fig 2E). sgSLC7A3- or control sgRen-expressing cells were seeded in a 1:1 ratio with cells co-expressing control sgRen and GFP, allowing us to monitor differential growth over time by flow cytometry. Validating the results obtained in the gof screen, SLC7A3-overexpressing cells showed increased fitness compared with control cells when

cultured in low arginine media, whereas no difference was observed in fully reconstituted media or when sgRen-expressing cells were used (Fig 2F). Interestingly, the competitive advantage of sgSLC7A3-bearing cells was stronger at the lowest concentration of arginine and enrichment increased over culture time and cell replating. To further confirm that this effect was mediated by on-target effects and compare the results obtained by the SAM-mediated up-regulation of SLC genes from the endogenous promoter with the ectopic, cDNA-driven overexpression of these same transporters, we established an analogous CCA using HEK293T cells lines expressing codon-optimized versions of SLC7A3 or GFP under the control of a doxycycline-inducible promoter. Treatment with doxycycline for 24 h resulted in a substantial increase of SLC7A3 protein levels (Fig S2E). We then monitored differential growth in CCA by co-culturing parental (empty) or SLC7A3 overexpressing lines together with GFP expressing cells. Supporting the results obtained with the SAM approach, SLC7A3-overexpressing cells showed increased fitness in low arginine-containing (1%) media but not in fully reconstituted media (Fig 2G). Together, these data support the validity of our gof approach to identify SLCs sustaining cell fitness upon nutrient deprivation and highlight SLC7A3 up-regulation as a strategy to resist arginine scarcity.

### A systematic screen for transporters overcoming extracellular amino acid limitation

Encouraged by these results, we extended this approach to systematically screen for SLCs sustaining cellular fitness under limiting concentrations of other amino acids. We therefore tested the effect on cell fitness of selectively reducing the concentration of each of the 15 amino acids present in DMEM culture media and observed a strong impairment of cell viability for all amino acids with the exception of glycine or serine, in line with the lack of significant transcriptional responses observed upon depletion of these two amino acids (Figs 1 and S1). Based on the arginine and SLC7A3 validation results (Fig 2F), we selected conditions with reduced concentration of amino acids imposing a strong selective pressure (between 0.1% and 2% of the normal DMEM concentration, Fig S3A) and performed the screen over 14 d including three passages where cells were detached and reseeded in fresh media (Fig 3A). SAM-HEK293T cells were transduced with the SLC-SAM library and cultivated in media with reduced levels of single amino acids or lacking all amino acids, whereas cells grown for the same period in media reconstituted with all amino acids served as control (Tables S4–S7). Fig 3B provides an overview of the screening outcome highlighting significantly enriched SLCs for each condition. Confirming the robustness of our approach, we could recapitulate in these settings the specific enrichment of SLC7A3 sgRNAs in arginine-reduced conditions (Fig 3B and C). Moreover, specific enrichment of SLC7A3 was also observed in lysine-reduced

SLC7A3 in limiting arginine media concentrations. Data shown derived from one experiment performed in three replicates. Results are representative of two independent experiments. Statistical significance at day 13 was calculated using one-way ANOVA with Dunnett's test. ***$P \leq 0.001$, **$P \leq 0.01$, *$P \leq 0.05$, ns, not significant. **(G)** CCA showing enrichment of HEK239 Flip-IN cells ectopically expressing C-terminally Strep-HA (SH)–tagged SLC7A3 compared with control HEK239 Flip-IN cells expressing eGFP in media conditions with limiting arginine concentrations (0.1%). Cells were treated with doxycycline for the duration of the assay. Data shown for two independent experiments each performed in triplicates. Statistical significance was calculated using the two-way ANOVA with Šidák's test. ***$P \leq 0.001$, **$P \leq 0.01$, *$P \leq 0.05$, ns, not significant.

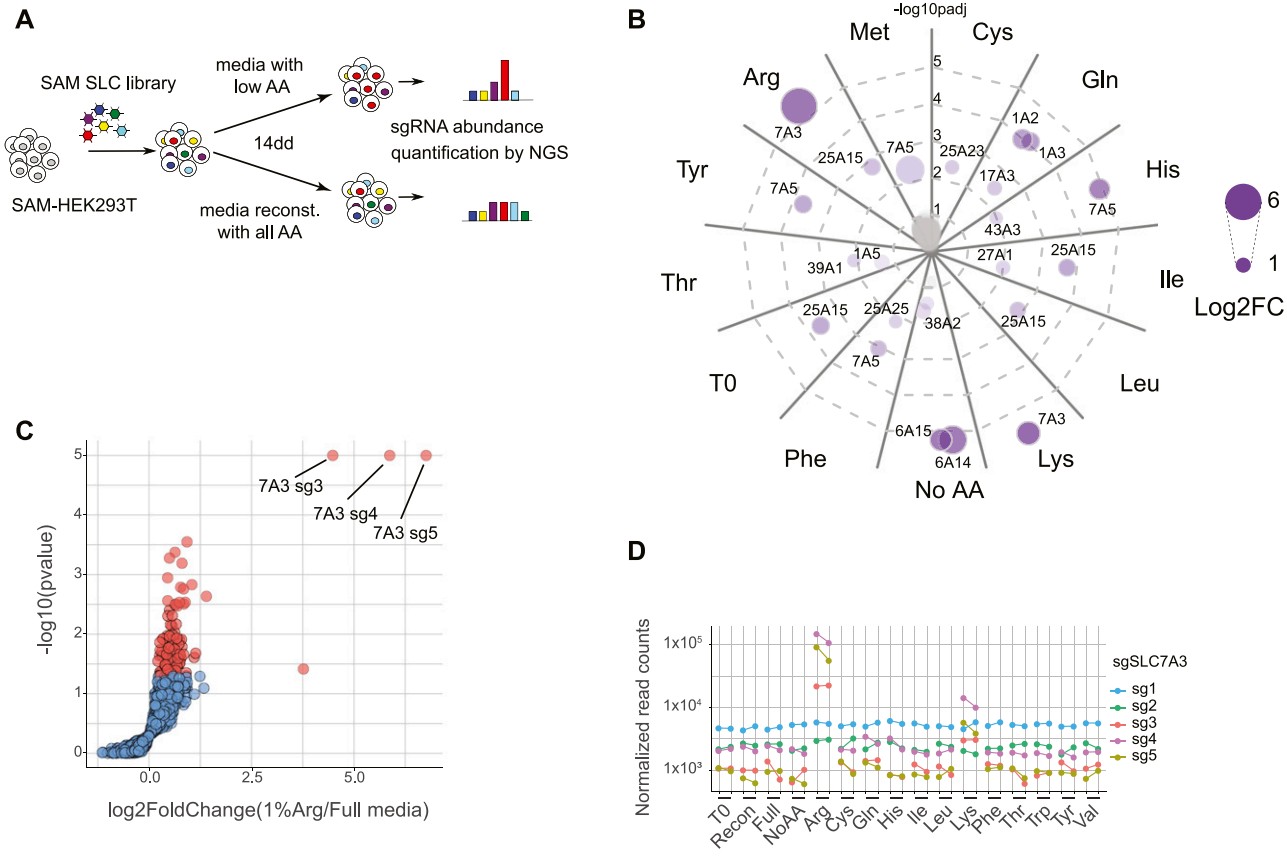

**Figure 3. Systematic identification of SLCs sustaining cellular fitness upon single–amino acid perturbations.**
**(A)** Experimental setup of the screens for SLCs able to overcome amino acid limitations in the extracellular media. **(B)** Overview of the significantly enriched SLCs for each of the tested conditions at FDR < 0.01. The circle size reflects the log$_2$ fold change above the fully reconstituted media, whereas the distance from the center and the intensity of color reflect the statistical significance of the effect observed. **(C)** Volcano plot showing the enrichment and statistical significance of each sgRNA upon arginine starvation. **(D)** Plot showing the normalized counts of SLC7A3-targeting sgRNAs across the conditions tested.

conditions (Fig 3B), a second amino acid described to be a substrate for this transporter (46). Indeed, monitoring *SLC7A3*-sgRNA abundance across all the tested conditions, we observed that three sgRNAs accumulated specifically in Arg/Lys-reduced media, whereas two additional sgRNAs did not, possibly because the latter were less efficient in triggering transcriptional activation (Fig 3D). In addition to *SLC7A3*, another member of the SLC7 family of amino acid transporters expressed at the plasma membrane, *SLC7A5*, was identified in multiple conditions (Figs 3B and S3B and E). SLC7A5 belongs to the second subgroup (after the CAT proteins) of the SLC7 family, the HATs (heterodimeric amino acid transporters), which work as obligate heterodimers with SLC3 family members and display differences in term of substrate specificity and mode of transport (46). These heteromeric amino acid transporters work mostly as exchangers, and SLC7A5/SLC3A2 substrates comprise large neutral amino acids which are translocated in a sodium- and pH-independent manner. *SLC7A5* was the most enriched gene in the screens for histidine, tyrosine, phenylalanine, and methionine (Figs 3B and S3B) and, with a less stringent sgRNA cutoff, also with isoleucine and leucine (Fig S3D), therefore recapitulating most of the known transporter substrates.

Analyzing the other screening conditions, we observed enrichment of the aspartate and glutamate transporters of the SLC1 family, *SLC1A2* (*EAAT2/GLT-1*) and *SLC1A3* (*EAAT1/GLAST-1*), upon glutamine starvation (Figs 3B and S3B, D, and E). This is consistent with recent reports showing that cytosolic aspartate concentrations determine cell survival upon glutamine starvation and that expression of SLC1A3 overcomes the effects of limiting glutamine availability (47, 48). We also detected enrichment for the mitochondrial proton-coupled ornithine and citrulline transporter *SLC25A15* (49) in several conditions (including full media and initial, time 0 condition), likely a result of lower sgRNA abundance in the fully reconstituted media condition (the control condition used to calculate the enrichment, Figs 3C and S3C and D). Finally, we observed an enrichment for the two broad-specificity, concentrative amino acid transporters *SLC6A14* and *SLC6A15* (44) when all amino acids were removed from the culture media, providing therefore conditions where the only amino acid sources were dialyzed serum or possibly dying cells (Figs 3C and S3B, D, and E). Overall, our gof screen identified key concentrative amino acid transporters able to overcome specific and general amino acid limitation in the extracellular medium.

### Identified SLCs sustain cellular fitness upon starvation of their amino acid substrates or via metabolic compensation

To further validate the specificity and characterize the role of the identified transporters upon amino acid limitation, we tested several of these SLCs using the above-described color competition assay. sg*SLC7A3*-expressing cells showed growth advantage in media containing low levels of the cationic amino acid arginine and lysine but not in case of the non-substrate histidine nor in reconstituted media (Fig 4A). We then investigated the role of the other SLC7 member showing enrichment in multiple conditions and confirmed up-regulation at the protein level of SLC7A5 by the three top-scoring sgRNAs identified in the screen (Fig 4B and C). Interestingly, sg*SLC7A5*-expressing cells showed a concomitant increase in SLC3A2 protein abundance, which correlated with SLC7A5 induction (Fig 4C). This suggests that up-regulated SLC7A5 assembles with its SLC3A2 partner in functional heterodimers and that proteostatic co-regulatory mechanisms control SLC3A2 in response to SLC7A5 levels as *SLC3A2* mRNA was largely unaltered in sg*SLC7A5*-expressing cells (Fig S4A). In CCA, *SLC7A5* transcriptional activation conferred advantage in histidine- and tyrosine- but not arginine-low media (Fig 4B). These results demonstrated that different SLC7 family members sustain cellular fitness in conditions where cells are starved from one of their amino acid substrates.

Finally, transduction of SAM-HEK293T cells with the top three sgRNAs targeting *SLC1A2* or *SLC1A3* resulted in increased protein levels of the two transporters (Fig S4B and C) and conferred a modest but consistent selective advantage in low-glutamine media (Fig 4D and E), therefore confirming the screen outcome. To further investigate this effect, we turned to the cDNA-mediated CCA and generated HEK293T cells lines expressing codon-optimized versions of *SLC1A2* and *SLC1A3*. Treatment with doxycycline strongly up-regulated transporter expression (Fig S4D–F), and immunofluorescence experiments confirmed plasma membrane localization of the overexpressed SLC1A2/3 proteins (Fig S4G and H). We then monitored differential growth by co-culturing parental (empty), SLC1A2, or SLC1A3 overexpressing lines together with GFP expressing cells. In line with the results obtained with the SAM approach, cells expressing SLC1A2 or SLC1A3 displayed increased fitness specifically in low glutamine but not in reconstituted media, irrespective of the presence of an N-terminal Strep-HA tag (Fig 4F and G). Interestingly, a similar fitness advantage was observed also in media completely devoid of glutamine, supporting the notion that SLC1A2 and SLC1A3 up-regulation leads to metabolic compensatory effects and not to direct glutamine uptake. Overall, these data validate the ability of a set of concentrative transporters to overcome amino acid limitations.

### A genome-wide gof screen supports *SLC7A3* as the prominent gene able to overcome extracellular arginine depletion

A question raised by the SLC-focused approach we developed and validated is how SLC up-regulation compares to other possible cellular mechanisms in sustaining cell viability under starvation conditions. The biased nature of the approach focused on SLC transporters, with all its advantages to functionally ascribe transporter-nutrient relationships, does not allow to gauge their relative

importance. We thus switched to a genome-wide perspective and performed a gof screen in arginine-limiting conditions using a library covering all protein-encoding gene products (Fig 5A). We infected SAM-HEK293T cells with a genome-wide SAM library carrying 70,290 sgRNAs targeting 23,430 genes (35) and cultivated them with 1% arginine or fully reconstituted media before collecting samples 14 d later. Comparison of the sgRNA abundances between the arginine-limited and reconstituted media showed a clear enrichment for multiple sgRNAs targeting SLC7A3 (Fig 5B and Tables S8 and S9). Accordingly, when analyzed at the gene level, *SLC7A3* was the single most-enriched gene identified, confirming the accuracy of our SLC-focused screens. Importantly, this result suggests that indeed, increased uptake of arginine by this transporter was the most effective mechanism to maintain growth-limiting amino acid homeostasis, at least under these experimental settings (Table S9). To gain a better mechanistic understanding of how SLC7A3 up-regulation promotes cell survival and support our hypothesis that this occurs by increasing the uptake of its substrate amino acids, we performed a targeted metabolomic analysis of SLC7A3-overexpressing cells (Figs 5C and S5A–C and Table S10). Rewardingly, this revealed that doxycycline-mediated up-regulation of SLC7A3 resulted in a striking and specific increase in the intracellular concentration of its substrates arginine, lysine and ornithine (46). Although most of the other metabolites were not affected, we observed an increase also of biogenic amines (symmetric dimethylarginine and α-aminoadipic acid), likely resulting from the elevated cellular levels of arginine and lysine, respectively.

These results highlight the prominent effect of transporter up-regulation in supporting the ability of cells to overcome limiting nutrient conditions by increasing their uptake.

## Discussion

The role of cellular metabolic responses to environmental changes and their effect on survival and proliferation in normal and pathological states is increasingly recognized (3, 50). Indeed, metabolic plasticity of human cells is an area of intense research as it is associated with a great variety of processes, ranging from T-cell activation (51) to metabolic symbiosis within the tumor microenvironment (52). The mechanisms by which human cells sense nutrient levels and adapt accordingly their metabolism are beginning to be elucidated and typically involve master regulators such as mTOR and AMPK, changes in transcription factors activity, and changes in the expression of rate-limiting enzymes and of membrane transporters (2, 3). Transporter (up)regulation is part of the cellular response to nutrient limitation, such as in low-glucose (53, 54) and low-glutamine (47, 48) conditions. Accordingly, several transporters are part of concerted transcriptional programs initiated by nutrient-sensing mechanisms (4, 55). What we wanted to clarify was if transporters are mere executing "workhorses" of metabolic adaptation or also capable, by their own action, to sustain metabolic corrections leading to cell state and cell fate decisions (e.g., growth, no growth). Another recurrent challenge resides in ascribing individual functions to transporters in light of

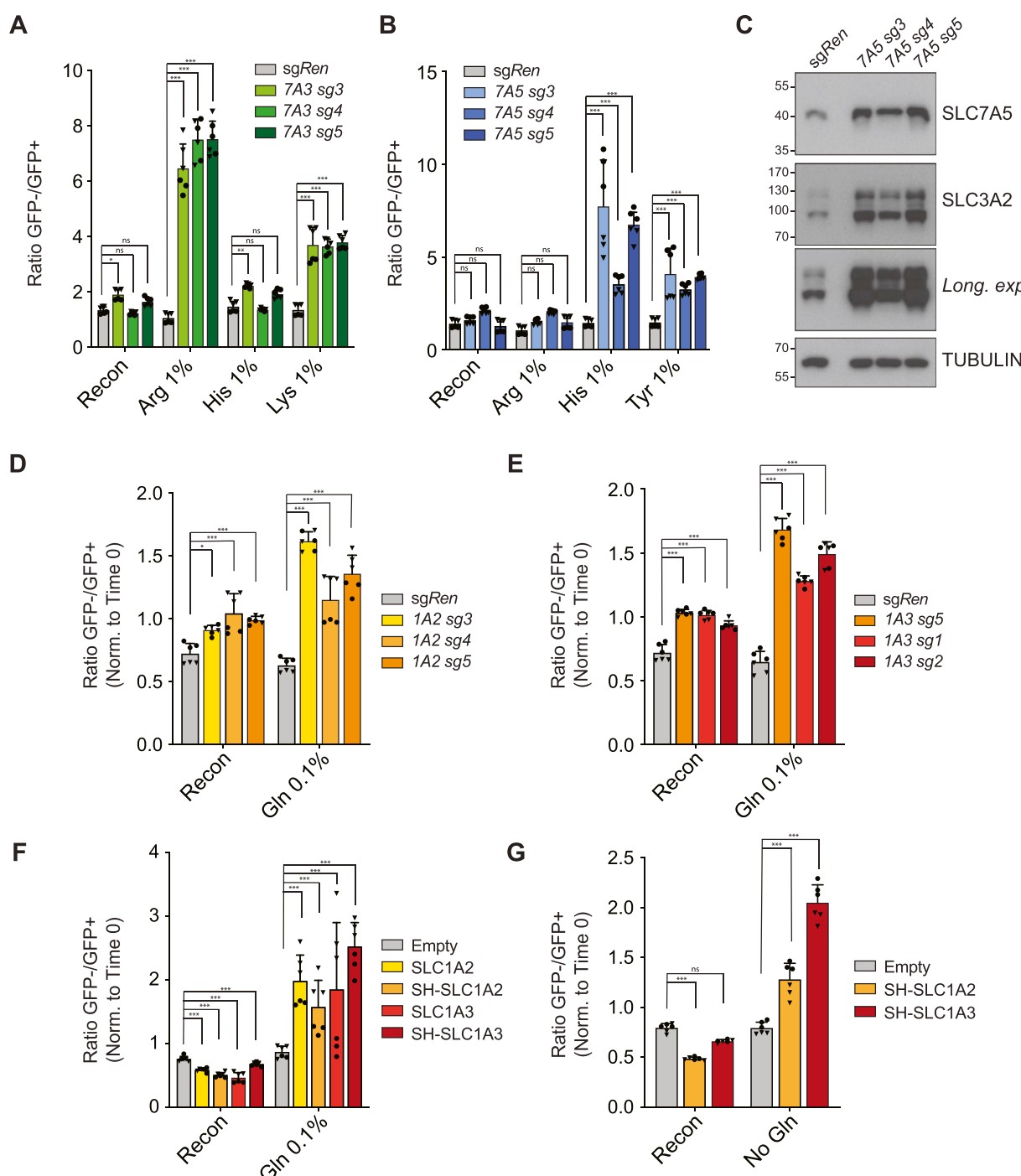

**Figure 4. Top screening hits confer selective fitness advantages in specific amino acid–restricted conditions.**
**(A)** Color competition assay (CCA) showing enrichment of eGFP-negative synergistic activation mediator (SAM)-HEK239T cells carrying sgRNA targeting *SLC7A3* or sg*Ren* compared with eGFP-positive sg*Ren* carrying control cells in media conditions with limiting concentrations of arginine and lysine but not histidine. Data shown for two independent experiments each performed in triplicates. **(B)** CCA showing enrichment of SAM-HEK239T cells carrying sgRNA targeting *SLC7A5* in media conditions with limiting concentrations of histidine and tyrosine but not arginine. Data shown for two independent experiments each performed in triplicates. **(C)** Western blot showing increased protein levels of SLC7A5 and the accessory chain SLC3A2 in SAM-HEK239T cells carrying sgRNA targeting *SLC7A5*. **(D, E)** CCA showing enrichment of SAM-HEK239T cells carrying sgRNA targeting *SLC1A2* (D) or *SLC1A3* (E) in media conditions with limiting concentrations of glutamine. Data shown for two independent experiments each performed in triplicates. **(F, G)** CCA showing enrichment of HEK239 Flip-IN cells ectopically expressing untagged or Strep-HA (SH)–tagged SLC1A2 or SLC1A3 compared with control HEK239 Flip-IN cells expressing eGFP in media conditions with limiting concentrations (0.1%) (F) or the absence (G) of glutamine. Cells were treated with doxycycline for the duration of the assay. Data shown for two independent experiments each performed in triplicates. Statistical significance was calculated using two-way ANOVA with Dunnett's test. ***$P \leq 0.001$, **$P \leq 0.01$, *$P \leq 0.05$, ns, not significant.

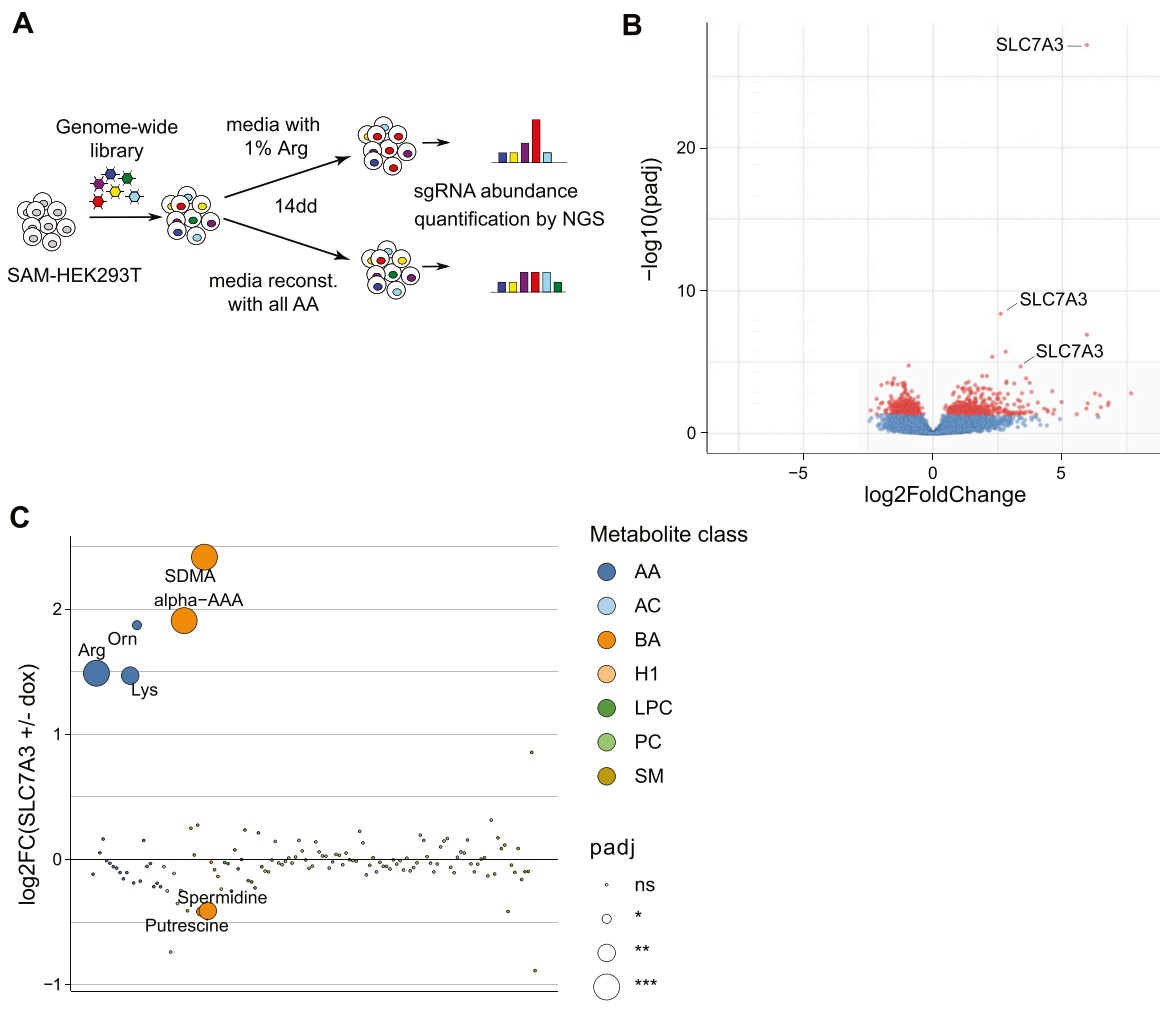

**Figure 5. Increased substrate uptake by SLC7A3 is a prominent gain-of-function mechanism to overcome arginine starvation.**
**(A)** Schematic representation of the experimental setup for the genome-wide synergistic activation mediator screen upon limiting arginine conditions. **(B)** Volcano plot showing sgRNA enrichment between low arginine and fully reconstituted conditions. **(C)** Targeted metabolomics analysis of doxycycline-induced HEK293 Jump-In cells overexpressing SLC7A3, compared with non-induced cells, shows enrichment for transporter substrates and related compounds. AA, amino acids; AC, acylcarnitines; BA, biogenic amines; H1, hexoses; LPC, lysophosphoatidylcholine; PC, phosphoatidylcholine; SM, sphingomyelin.

the high level of functional redundancy and plasticity in occurring expression patterns.

The transcriptional responses induced by depletion of individual amino acids in HEK293T and HeLa cells were largely independent of the specific amino acid depleted and included overexpression of a conserved set of amino acid transporters. Several of the well-characterized amino acid transporters families, including the SLC1, SLC7, and SLC38 solute carrier subfamilies, were well represented.

Using a highly scalable SLC-focused CRISPR/Cas9–based approach we developed, we identified both cases of direct uptake of the limiting nutrient by the enriched transporter and cases where the transporter overexpression is likely to rescue the nutrient limitation indirectly by providing metabolically related compounds. As examples of the first case, we identified two members of the SLC7 family, SLC7A3 (CAT3) and SLC7A5 (LAT1), as the main transporters able to restore cell fitness upon low extracellular concentration of arginine and lysine, for SLC7A3, and multiple amino acids including histidine, tyrosine, and methionine for SLC7A5. We were therefore

able to identify, starting from a completely unbiased approach, the known substrates for both transporters, highlighting the potential of this approach for substrate/transporter associations and functional annotation of uncharacterized transporters. Importantly, overexpression of SLC7A3 (CAT3) was confirmed in a genome-wide gof screen as the top case of single gene overexpression able to overcome arginine limitation. This was quite surprising and revealed a perhaps underestimated role of transporters upon environmental limitations. Exemplifying the second scenario, we identified the aspartate/glutamate transporters SLC1A2 and SLC1A3 as the two solute carriers able to sustain cellular fitness in media containing low or no glutamine. Glutamine is the most abundant free amino acid in cells, acting as biosynthetic substrate and exerting a critical energetic role by providing intermediates for the TCA cycle. Recent studies demonstrated that aspartate uptake is able to rescue cells from limiting glutamine conditions (47, 48). SLC1A2/3 up-regulation could therefore act by sustaining intra-cellular levels of aspartate and/or glutamate, which can feed the

TCA cycle. Because DMEM media does not contain aspartate and glutamate, it is likely that these may derive from dying cells. Further studies will be needed to investigate this hypothesis.

Although these cases refer to specific amino acid conditions, the finding that the broad-substrate amino acid transporters *SLC6A14* and *SLC6A15* were enriched specifically upon low extracellular concentrations of all amino acids is intriguing. One possibility is that these transporters are particularly energetically expensive as they couple amino acid uptake to the transport of both sodium and chloride ions. Considering the growing interest in SLC6A14 (20,21,56) and amino acid transporters in general as a potential cancer target, these results warrant further studies to understand the effect of environmental conditions on transporters expression.

Solute carriers represent the largest family of transporters in humans and a relatively understudied family, with ~30% of its members still lacking endogenous substrates despite an increasing recognition of their potential as drug targets and major roles in metabolism (25, 37). The approach described here allows for the systematic interrogation of the solute carrier family to identify the transporter(s) able to rescue nutrient-limiting conditions, providing insights into the range and potency of the responses available to cells in challenging environments. Large-scale SLC-focused gof screens may thus represent a powerful and efficient approach to identify the relevant transporters by their function and likely contribute to the ongoing efforts of de-orphanization (57) by overcoming the issue of functional redundancy. An obvious limitation of the strategy concerns those metabolites that are not rate-limiting for growth and would require the development of specific screening read-outs. Another potential shortcoming is the possibility that, when overexpressed, transporters may lose activity or selectivity because of the absence of a partner or a modification.

Despite these limitations, the results presented here provide compelling data to support the hypothesis that up-regulation of individual SLCs membrane transporters alone can be sufficient to overcome metabolic bottlenecks. Transporters, alone and in combinations, may thus exert a more central role than generally estimated in governing the metabolic plasticity required for cellular and tissue homeostasis. Because of their role in mediating the cellular needs with the environment, they may be ideally suited to act as conveyors of metabolic integration across tissues. Their dynamic mRNA and protein expression profiles may in future be used to interpret metabolic states of cells and tissues. Altogether, the role in overcoming metabolic bottlenecks increases their attractivity as potential drug targets. In this sense, the approach presented here adds another assay for the identification of targeting chemotherapeutics.

# Materials and Methods

## Cell lines and reagents

HEK293T were obtained from ATCC, HeLa were provided by M Hentze. HEK293 Jump-In and Flp-In TREx cells that allow doxycycline-dependent transgene expression were from Invitrogen. Cell lines were checked for mycoplasma infection by PCR or ELISA. Cell lines were authenticated by STR profiling. Cells were kept in DMEM (Gibco or Sigma-Aldrich) supplemented with 10% (vol/vol) FBS (Gibco) and antibiotics (100 U/ml penicillin and 100 mg/ml streptomycin, Gibco or Sigma-Aldrich) and shifted to specific nutrient-restricted media for the different assay (described below).

## Antibodies

Antibodies used were SLC7A3 (HPA003629; Sigma-Aldrich), SLC7A5 (#5347; Cell Signaling), SLC3A2 (376815, E-5; Santa Cruz), tubulin (Ab-7291; Abcam), HA (MMS-101P; Covance), actin (AAN01-A; Cytoskeleton), SLC1A2 (365634; Santa Cruz), SLC1A3 (#5684T; Cell Signaling), and HRP-conjugated secondary antibodies (Jackson Immunoresearch).

## Plasmids and cell line generation

SgRNA were cloned into lenti sgRNA(MS2)_zeo backbone (Addgene: 61427). Cloned oligonucleotides, named accordingly to the SLC SAM library ID, were as follows: *SLC7A3* (sg3 F caccgGGCTTTGCAAAAGGATTGCG, R aaacCGCAATCCTTTTGCAAAGCCc; sg4 F caccgTGAGGATGGGACGCAGTCTC, R aaacGAGACTGCGTCCCATCCTCAc; sg5 F caccgTAGCGAGGAGGATTGGGGGT, R aaacACCCCCAATCCTCCTCGCTAc); *SLC7A5* (sg3 F caccgCCCGCCCCCTCGGCC-CAGCT, R aaacAGCTGGGCCGAGGGGGCGGGc; sg4 F caccgGAGG-GACGGGGCCGGGCCAC, R aaacGTGGCCCGGCCCCGTCCCTCc; sg5 F caccgGTGCGTCGTCCGGCCCAGCC, R aaacGGCTGGGCCGGACGACGCACc), *SLC1A2* (sg3 F caccgAGATCCTGGGCTCCTGCCAC, R aaacGTGG-CAGGAGCCCAGGATCTc; sg4 F caccgGGCAGAGGAGGGACCGCGAC, R aaacGTCGCGGTCCCTCCTCTGCCc; sg5 F caccgAAAGGAGTTGCCC-GAGGCGG, R aaacCCGCCTCGGGCAACTCCTTTc), *sgSLC1A3* (sg1 F caccg-CACCCTCGTCTTCCCTGAAA, R aaacTTTCAGGGAAGACGAGGGTGc; sg2 F caccgAGGAAACATGCAATAATGTG, R aaacCACATTATTGCATGTTTCCTc; sg5 F caccgCTCGTAACAGTTGTACAACC, R aaacGGTTGTACAACTGTTACGAGc). sgRen targeting *Renilla* luciferase cDNA was used as negative-control sgRNA (GGTATAATACACCGCGCTAC) (33) and cloned into lenti sgRNA(MS2)-zeo backbone (Addgene: 61427) or lenti sgRNA(MS2)-zeo-IRES-eGFP backbone. Codon-optimized cDNAs for human *SLC7A3*, *SLC1A2*, and *SLC1A3* were obtained from Genscript and cloned in pTO vectors with or without a N- or C-terminal Strep-HA tag.

## Transcriptomic analysis of cells under nutrient-limiting conditions

HEK293T and HeLa cells were seeded in full media (DMEM Gibco, 10% FBS, antibiotics). After 24 h, media was removed and, after a wash with PBS, substituted with DMEM media lacking the indicated amino acid supplemented with 10% dialyzed FBS (Cat. no. 26400-044; Gibco). Starvation media lacking a specific amino acid were prepared by complementing amino acid–free DMEM media (i.e., devoid of all 15 amino acids normally present, custom made by PAN biotech) with the other 14 amino acids (from individual powders; Sigma-Aldrich). DMEM media reconstituted with all 15 amino acids and 10% dialyzed FBS as well as full media served as controls. For HEK293T cells, the following additional conditions were tested: 100 nM Torin-1 (Tocris Bioscience) in DMEM media reconstituted with all 15 amino acids and 10% dialyzed FBS; DMEM without vitamins (custom made by PAN biotech) and 10% dialyzed FBS; and DMEM without glucose and pyruvate (custom made by PAN biotech) and

10% dialyzed FBS. After 16 h, media was removed and cells were harvested in cold PBS. Total RNA was isolated using the QIAGEN RNeasy Mini kit including a DNase I digest step. RNA-sequencing (RNA-seq) libraries were prepared using QuantSeq 3′ mRNA-Seq Library Prep Kit FWD for Illumina (Lexogen) according to the manufacturer's protocol. Libraries were subjected to 50-bp single-end high-throughput sequencing on an Illumina HiSeq 4000 platform at the Biomedical Sequencing Facility (https://biomedical-sequencing.at/). Raw sequencing reads were demultiplexed, and after barcode, adapter and quality trimming with cutadapt (https://cutadapt.readthedocs.io/en/stable/), quality control was performed using FastQC (http://www.bioinformatics.babraham.ac.uk/projects/fastqc/). The remaining reads were mapped to the GRCh38 (h38) human genome assembly using genomic short-read RNA-Seq aligner STAR version 2.6.1as (58). We obtained more than 98% mapped reads in each sample with 70–80% of reads mapping to unique genomic location. Transcripts were quantified using the End Sequence Analysis Toolkit (ESAT) (59). Differential expression analysis was performed using three biological replicates with DESeq2 (1.21.21) on the basis of read counts (60). Exploratory data analysis and visualizations were performed in R-project version 3.4.2 (Foundation for Statistical Computing, https://www.R-project.org/) with Rstudio IDE version 1.0.143, ggplot2 (3.0.0), dplyr (0.7.6), readr (1.1.1), gplots (3.0.1).

## Generation of a SLC-focused gof library

A set of sgRNAs targeting 388 human SLC genes, with up to six sgRNA per RefSeq transcript, were generated using the Cas9 activator tool (34). An additional set of 120 non-targeting sgRNAs was included by generating random 20-mers and selecting for sequences with at least three mismatches from any genomic sequence. Adapter sequences were added to the 5′ and 3′ sequences (5′ prefix: TGGAAAG-GACGAAACACCG, 3′ suffix: GTTTTAGAGCTAGGCCAACATGAGGAT) to allow cloning by Gibson assembly in the lenti-sgRNA(MS2)_EF1A-zeo cloning backbone (#61427; Addgene). The oligos were synthetized as a pool by LC Sciences. Full-length oligonucleotides (66 nt) were amplified by PCR using Phusion HS Flex (NEB) and size-selected using a 2% agarose gel (Primers: SAM_ArrayF TAACTTGAAAGTATTTCGATTTCTTGGCTTTATA-TATCTTGTGGAAAGGAC GAAACACCG, SAM_ArrayR TTTTAACTTGC-TAGGCCCTGCAGACATGGGTGATCCTCATGTTGG CCTAGCTCTAAAAC).

The vector was digested with BsmBI (NEB) for 1 h at 55°C, heat inactivated for 20′ at 80°C, followed by incubation with antarctic phosphatase (NEB) for 30′ at 37°C. A 10 µl Gibson ligation reaction (NEB) was performed using 5 ng of the gel-purified inserts and 12.5 ng of the vector, incubated for 1 h at 50°C, and dialyzed against water for 1 h at RT. The reaction was then transformed in Lucigen Endura cells and plated on two 245-mm plates. Colonies were grown at 32°C for 16–20 h and then scraped from the plates. The plasmid was purified with the Endo-Free Mega prep kit (QIAGEN) and sequenced by NGS using the approach described in Konermann et al (35). The library is available on Addgene (#132561).

## Genetic screening in limiting nutrient conditions

HEK293T cells were used to generate lentiviral particles by transient transfection of lentiviral constructs (SLC-SAM library or genome-wide SAM library #1000000057; Addgene) and packaging plasmids

psPAX2 (#12260; Addgene), pMD2.G (#12259; Addgene) using PolyFect (QIAGEN). After 24 h, the medium was changed to fresh IMDM or DMEM medium supplemented with 10% FCS and antibiotics. Viral supernatants were collected after 48 h, filtered, and stored at −80°C until further use. To generate the cells containing the SAM machinery, HEK293T cells were then infected with viral particles carrying a catalytically dead Cas9-VP64 (#61425; Addgene) and MS2-P65-HSF1 constructs (#89308; Addgene). For the genetic screening, dilution factors for library transduction at a MOI of 0.2–0.3 were determined by zeocin survival after transduction. For SLC SAM–based screens (Fig 3), cells were infected with the SLC SAM library at high coverage (>1,000×) and, after selection for 7 d with zeocin (250 µg ml$^{-1}$), transferred in media containing modified amino acid concentrations, as listed in Fig S3A (5 × 10$^6$ cells/condition, two replicates/condition). Cells in amino acid–depleted conditions were passaged every 3 d and cells in full and reconstituted media when close to confluency. After 14 d, the cells were collected, washed with PBS, and stored at −80°C until further use. Cells grown in low-methionine media for 14 d were amplified for 5 d in full media before collection. Cells grown in media lacking all amino acids for 12 d were amplified for 5 d in full media before collection. For genome-wide SAM-based screens (Fig 5), cells were infected with the genome-wide SAM library at high coverage (1,000×) and, after selection for 10 d with zeocin (250 µg ml$^{-1}$), transferred in media containing either low-arginine concentrations (1%) or fully reconstituted supplemented with 10% dialyzed FBS (Cat. no. 26400-044; Gibco) (50 × 10$^6$ cells/condition, three replicates/condition). Cells in arginine-depleted media were passaged every 3 d and cells in reconstituted media when close to confluency. After 14 d, the cells were collected, washed with PBS, and stored at −80°C until further use. For all screens, genomic DNA was extracted using the DNAeasy kit (QIAGEN), and the sgRNA cassette amplified by PCR as described in reference 33.

## Screen analysis and hit scoring

sgRNA sequences were extracted from NGS reads, matched against the original sgRNA library index, and counted using an in-house Python script. Differential abundance of individual sgRNAs was estimated using DESeq2 (1.21.21) (60). Contrasts were performed individually for each treatment and significance was tested using either one- or two-tailed Wald tests. The sgRNAs were then sorted by P-value and aggregated into genes using GSEA (fgsea R package v1.7). To avoid false positives, only significant sgRNAs ($P \leq 0.05$) were considered for enrichment, requiring also a minimum of two sgRNAs per gene. Gene enrichment significance was estimated by a permutation test using 10$^8$ permutations, and P-values were corrected for multiple testing using the Benjamini–Hochberg procedure (FDR) (61).

## Flow cytometry–based competition assay

Flow cytometry–based color competition assay were performed using either the SAM component expressing HEK293T stably transduced with the indicated sgRNA in lenti sgRNA(MS2)-zeo (Addgene: 61427, eGFP negative) and control sgRen in lenti sgRNA(MS2)-zeo-IRES-eGFP (eGFP positive) or, for cDNA-based overexpression

experiments, using HEK293 FlipIN empty cells (parental) or expressing the indicated SLCs and control HEK293 FlipIN cells expressing eGFP. The cell populations were mixed in a 1:1 ratio and seeded in three replicates in full media. For cDNA-based over-expression CCA, doxycycline (1 μg/ml) was added when cells were mixed and kept for all the duration of the assay. After 24 h (day 0), full media was removed and, after a wash with PBS, exchanged with the indicated media. Every 3 d, cells were detached and reseeded into fresh indicated media for the duration of the assay (12 d or as indicated) and analyzed by flow cytometry.

The respective percentage of viable (FSC/SSC) eGFP-positive and eGFP-negative cells at the indicated time points was quantified by flow cytometry and ratio (eGFP–/eGFP+) were calculated. Where indicated, ratios were normalized to the ratio (eGFP–/eGFP+) at day 0.

## Immunofluorescence

SLC1A2/A3-expressing HEK293 Flip-In cells were plated on poly-L-lysine hydrobromide (Sigma-Aldrich)–coated glass coverslips and induced with doxycycline (1 μg/ml) where indicated. After 24 h, cells were washed with PBS, fixed (PBS, 2% formaldehyde), and per-meabilized (PBS, 0.3% saposin, 10% FBS). Slides were incubated with anti-HA (MMS-101P; Covance) and anti-AIF (5318; Cell Signaling) antibodies (1 h, RT, PBS, 0.3% saposin, 10% FBS). After three washes, slides were incubated with Alexa-Fluor-488–coupled anti-mouse (A11001; Life Technologies) and Alexa-Fluor-594–coupled anti-rabbit (A11012; Life Technologies) (1 h, RT, PBS, 0.3% saposin, 10% FBS). After DAPI staining, slides were washed three times and mounted on coverslips with ProLong Gold (Invitrogen). Images were taken with a Zeiss Laser Scanning Microscope (LSM) 700.

## Q-PCR

Total RNA was isolated using the RNeasy Mini Kit (QIAGEN) including DNase I digestion step. The reverse transcription was performed using the RevertAid First Strand cDNA Synthesis Kit (Thermo Fisher Scientific) using oligo dT primers. Quantitative PCRs were carried out on Rotor Gene Q (QIAGEN) PCR machine using the SensiMix SYBR kit (Bioline) or on Roche LightCycler 480 II using Luna universal qPCR master mix (NEB). Results were quantified using the $2^{-\Delta\Delta Ct}$ method, using GAPDH or HPRT as the reference. The primers used were: *GAPDH* F: 5′-CCTGACCTGCCGTCTAGAAA-3′, R: 5′-CTCCGACGCCTGCTTCAC-3′; *SLC7A3* F: 5′-AACTCGGCTTAACTCCGCCT-3′, R: 5′-CACCTCGCCAGCTAGGACAT-3′; *HPRT* F: 5′-AGACTTTGCTTTCCTTGGTCAG-3′, R: 5′-CCAACAAAGTCTGGCTTATATCC-3′; *SLC3A2* F: 5′-GACTTCCTTCTTGCCGGCTC-3′, R: 5′-AGGGAAGCTG-GACTCATCCC-3′.

## Immunoblot

Cells were lysed in E1A lysis buffer (50 mM Hepes, 250 mM NaCl, 5 mM EDTA, 1% NP-40, pH 7.4) supplemented with EDTA-free protease in-hibitor cocktail (1 tablet per 50 ml; Roche) and Halt phosphatase in-hibitor cocktail (Thermo Fisher Scientific) for 10 min on ice. Lysates were cleared by centrifugation at 18,000g, 10 min, 4°C, and normalized by BCA (Thermo Fisher Scientific) using BSA as the standard. Where indicated, cleared lysate was incubated without or with PNGase F (NEB, 1 μl per 40 μl lysate) for 30 min at 37°C. Typically, 10-20 μg protein per

sample was resolved by SDS–PAGE and blotted to nitrocellulose membranes. Membranes were blocked with 5% nonfat dry milk in TBST, probed with the indicated antibodies, and detected with horseradish peroxidase–conjugated secondary antibodies.

## Metabolomics

The HEK293 Jump-In TREx SLC7A3 cell line was plated at sub-confluent density in a cell culture–treated six-well plate in nor-mal growth medium with or without 1 μg/ml doxycycline (six replicates/condition, 750,000 cells/well). Cells were incubated overnight at 37°C, 21% $O_2$, in a humidified incubator. The following day, cells were washed gently two times with 500 μl PBS (room temperature). The plate was transferred to ice, and 1,500 μl/well of 80:20 ice-cold MeOH:$H_2O$ solution was added; cells were scraped and transferred to pre-cooled Eppendorf tubes. Samples were shortly vortexed, snap frozen in liquid nitrogen, thawed on ice, and centrifuged at 20,800g at 4°C for 10 min. Supernatant was trans-ferred into a HPLC vial and evaporated using a nitrogen evaporator. Evaporated samples were reconstituted in 50 μl of 80:20 MeOH:$H_2O$ and stored at –80°C until analysis. The targeted metabolomics was performed using the Biocrates Absolute IDQ p180 kit. The kit pro-vides (semi)quantitative analysis of 184 metabolites. Sample preparation was performed according to the User Manual provided by the manu-facturer. Briefly, 10 μl of internal standard was loaded onto the 96-well kit plate, dried under nitrogen, and followed by loading 10 μl of samples, blanks, calibration standards, and quality-control samples. After an ad-ditional drying step, derivatization reagent was added, and samples were incubated for 20 min at room temperature. The 96-well plate was dried, and the extraction solvent was added. Supernatants were collected in a new 96-well kit plate by centrifugation, diluted, and then analyzed using a Water Acquity UHPLC system and Waters Xevo TQ-MS mass spectrometer. Each sample was analyzed in using two methods: (1) metabolites were measured with the LC-MS/MS method (positive ionization mode, sample injected using a HPLC column) and (2) lipids were measured with the FIA-MS/MS method (positive mode, sample injected without a HPLC column). The UHPLC method files KIT2_LC_8015.IPR (.w2200, .wvhp) and KIT3-FIA_8015.IPR (.w2200, .wvhp) are part of the kit provided by Biocrates. The acquired LC-MS/MS data were processed with Waters MassLynx V4.1 software TargetLynx module using the processing method KIT2_-WatersQuan_8015.mdb provided with the kit by Biocrates. The obtained results were imported in Biocrates METIDQ software where further validation of calibration standards and quality-control sample accuracy as well as internal standard intensity were carried out. The acquired raw FIA-MS/MS data were directly imported into the Biocrates METIDQ software where it was processed and validated. The final results were exported as tsv files. Conditions were compared using Welch's *t* test, and *P*-value was subsequently corrected for multiple testing according to the Benjamini and Hochberg procedure (61).

# Data Availability

Genomics datasets are provided in Tables S3–S8. The SLC SAM library is available on Addgene (132561; Addgene). The transcriptomics datasets are deposited at GEO (GSE153034 and GSE199252). Metabolomics data

have been deposited at MetaboLights (MTBLS4268) and are available until release at https://datagsf.blob.core.windows.net/public/MTBLS4268.tar.gz.

## Supplementary Information

## Acknowledgements

We would like to acknowledge the members of the Superti-Furga group for critical discussions and suggestions. We thank Johannes W Bigenzahn for kindly providing critical reagents, Svenja Onstein for experimental support, and Ulrich Goldmann for computational support. Part of this work was carried out within the RESOLUTE project. This project has received funding from the Innovative Medicines Initiative 2 Joint Undertaking (JU) under grant agreement No 777372. The JU receives support from the European Union's Horizon 2020 research and innovation programme and EFPIA. This work was supported by the Austrian Academy of Sciences and the European Research Council Advanced Grant 695214 GameofGates.

### Author Contributions

M Rebsamen: conceptualization, data curation, formal analysis, supervision, validation, investigation, visualization, methodology, and writing—original draft, review, and editing.
E Girardi: formal analysis, investigation, visualization, methodology, and writing—original draft, review, and editing.
V Sedlyarov: data curation, software, formal analysis, visualization, and methodology.
S Scorzoni: validation and investigation.
K Papakostas: investigation.
M Vollert: investigation.
J Konecka: methodology.
B Guertl: investigation.
K Klavins: investigation.
T Wiedmer: investigation.
G Superti-Furga: conceptualization, supervision, funding acquisition, project administration, and writing—review and editing.

### Conflict of Interest Statement

The authors declare the following financial interests/personal relationships which may be considered as potential competing interests: EGirardi is currently an employee of the SLC-focused company Solgate GmbH. G Superti-Furga is a co-author of patent applications related to SLCs, co-founder of Solgate GmbH, and the academic project coordinator of the IMI grants RESOLUTE and Resolution in partnership with Pfizer, Novartis, Bayer, Sanofi, Boehringer-Ingelheim, and Vifor Pharma. The G Superti-Furga laboratory receives funds from Pfizer.

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
