## [Reviewer comments · Life Science Alliance]

Life Science Alliance

Gain-of-function genetic screens identify SLC transporters overcoming nutrient restrictions

Giulio Superti-Furga, Manuele Rebsamen, Enrico Girardi, Vitaly Sedlyarov, Stefania Scorzoni, Konstantinos Papakostas, Manuela Vollert, Justyna Konecka, Bettina Guertl, Kristaps Klavins, and Tabea Wiedmer

DOI: <https://doi.org/10.26508/lsa.202201404>

Corresponding author(s): Giulio Superti-Furga, CeMM Research Center for Molecular Medicine and Manuele Rebsamen, University of Lausanne

Review Timeline:

Submission Date:	2022-02-10
Editorial Decision:	2022-03-14
Revision Received:	2022-07-20
Editorial Decision:	2022-08-11
Revision Received:	2022-08-24
Accepted:	2022-08-25

Scientific Editor: Novella Guidi

Transaction Report:

March 14, 2022

Re: Life Science Alliance manuscript #LSA-2022-01404

Prof. Giulio Superti-Furga
CeMM Research Center for Molecular Medicine
CeMM Research Center for Molecular Medicine of the Austrian Academy of Sciences
Lazarettgasse 14
Vienna 1090
Austria

Dear Dr. Superti-Furga,

Thank you for submitting your manuscript entitled "Gain-of-function genetic screens identify SLC transporters overcoming nutrient restrictions" to Life Science Alliance. The manuscript was assessed by expert reviewers, whose comments are appended to this letter. We, thus, encourage you to submit a revised version of the manuscript back to LSA that responds to all the reviewers' points.

Thank you for this interesting contribution to Life Science Alliance. We are looking forward to receiving your revised manuscript.

Sincerely,

B. MANUSCRIPT ORGANIZATION AND FORMATTING:

Reviewer #1 (Comments to the Authors (Required)):

This manuscript describes an innovative gain of function screen to gain insight on the role of amino acid transporters. The authors show that SLC7A3 can overcome deficiencies in availability of both arginine and lysine in the environment. They also show that SLC7A5 can compensate for deficiencies of several neutral amino acids. Finally they show that SLC1A2/3 can compensate for deficiency of glutamine. This approach is useful to identify potential drug targets, for example in tumor cells, or to reveal the substrate specificity of transport proteins.

The experiments are well-controlled and clearly described. The paper is very well written. My main criticism relates to Figure 5. This experiment is described essentially as a confirmation of previous experiments. Yet, Fig 5b suggests that new insights on other compensatory pathways relating to arginine biogenesis may exist. Did you follow up on some of the other genes that were significantly up or down regulated in 5b? Could they be confirmed by dox induced overexpression as was done for SLC7A3? What about performing metabolic analysis for some?

Minor: page 10, did you mean 70,290 sgRNAs?

Reviewer #2 (Comments to the Authors (Required)):

In this manuscript titled "Gain-of-function genetic screens in human cells identify SLC transporters overcoming environmental nutrient restrictions", Rebsamen et al. describe functional relationships between membrane transporters and amino acid limitations. They performed transcriptomics analyses and CRISPRa screens to identify transporter that can overcome specific amino acid restrictions. While approach and findings are of interest to amino acid metabolism field, some critical points remain to be addressed.

Major points:

- While authors present qPCR data for mRNA level increases in SLC7A3 activation, membrane targeting and expression of SLC family membranes are also regulated at the protein level. Thus, activation of SLC7A3 expression by CRISPRa sgRNAs need to be shown at the protein level as well.
- In Fig 2e, authors provide a competition assay for validation of SLC7A3's rescue effect on Arg restriction. They used CRISPRa sgRNAs for validation; however, these sgRNAs may activate downstream genes and have pleiotropic effects. That's why, cDNA overexpression of SLC7A3 should be included to show specific rescue effects in Arg restriction.
- In Fig 2, authors perform GFP-based competition assays for validation. And, they nicely show that SLC7A3 confers competitive advantage. Does SLC7A3 overexpression also restore proliferative capacity of cells in Arg restriction back to complete media levels? How do SLC7A3 sgRNA cells grown in 1% Arg compare to those in reconstituted media.
- In the Results section, while describing Fig 3b, authors state "cytosolic aspartate concentrations determine cell survival upon glutamine starvation and that expression of SLC1A3 overcomes the effects of limiting glutamine availability". However, SLC1A3 can also transport Glutamate, that can bypass the requirement of Gln and directly feed into TCA cycle. Authors need to consider this possibility as well.

Minor points:

- P-values should be added to the following graphs: Fig 2e, 4a, 4b, 4d, 4e, 4f, 4g
- Can authors provide FACS data for color competition assays with a viability marker (DAPI, PI etc.) to show whether percentage of dead/live cell ratios over time in aa-restricted media?

Reviewer #3 (Comments to the Authors (Required)):

General:

This work describes and validates the development of a genetic screen to identify SLC transporters, the upregulation of which is involved in supporting growth upon depletion of specific amino acids from the growth medium of HEK293 cells. Specifically, SLC7A3 is identified as upregulated under arginine and lysine depletion and able to overcome the growth-limitations induced by these conditions. Also SLC7A5 and SLC1A2 and -A3 were shown to support growth under nutrient-limiting conditions. The screen, which is based on CRISPR/Cas9-based transcriptional activation of a library of SLCs from their endogenous loci, is shown to clearly have the strength to identify novel roles of these key transporters, as long as their substrates are limiting for growth.

Collectively, this is a very well-written manuscript describing a technically powerful study describing a method that will be potentially extremely useful to future studies of SLCs. My only regret is that the work raises numerous intriguing questions that are not addressed, such as the mechanisms of upregulation of each transporter as well as the normal physiological roles and interactions of the transporters. But the authors are well aware of this and do not claim otherwise - the manuscript is mainly intended to describe this elegant tool for the community. Thus, I have by far mostly praise for the work, but some queries and suggestions are provided below.

Specific:

p. 3. As mentioned in the introduction, autophagy is powerfully regulated by starvation, but in the rest of the manuscript, this is not further mentioned, and the transporters identified are mainly plasma membrane localized. It would be interesting to describe to what extent a role for SLCs localized to autophagosomes/lysosomes was observed? Perhaps it would be worth providing an overview, graphically or otherwise, of which organelles, the identified transporters localized to?

p. 5. "To identify the transcriptional programs induced by nutrient limitation..." is perhaps a bit exaggerated, since the authors actually do not go beyond the TF analysis in Fig 1c, and the enrichment of ATF4 was fully expected from the literature. Hence, I suggest to rephrase. It would in my view also be interesting to comment on the apparently unique TF enrichment profile after Ile starvation (with much less response for instance after starvation for all AAs, which is offhand surprising, at least to me) and perhaps on the strong C/EBP enrichment after Cys starvation.

Fig. 1d:

1) Isn't it surprising (or am I just missing something) that SLC6A14 and -15, which are so strongly enriched in this condition in the CRISPR screen, are not among the transporters upregulated when all amino acids are depleted in the mRNA sequencing analysis? Please discuss.

2) I know that this is not the goal of this work, but it would perhaps be worth also commenting on the many strongly DOWN-regulated SLCs, which also seem to be shared among many of the starvation conditions - do they make physiological sense, given what is known about them in the literature? And on another, to me surprising note: why is the methionine transporter SLC43A2 so strongly downregulated uniquely when all AAs are depleted?

P 9/ Fig. 3b: As stated by the authors the mitochondrial ornithine and citrulline transporter SLC25A15 is, interestingly, upregulated in many of the conditions tested - but I do not understand the argument for why that is so ("as a result of sgRNA depletion in the fully reconstituted media condition" - please rephrase - and if you can speculate on the physiological logic of this upregulation that would be very nice.

Fig 3c-d vs Fig 4a:

When only sg3, -4, and -5 are enriched (Fig 3c-d), possibly (as suggested by the authors) because the other sg's were less efficient in triggering transcriptional activation, why are sg1-, -2, and -3 used in the color competition assay, and how does the above interpretation fit with the fact that they do increase cell fitness?

P 10 vs p 13. The explanations offered for the upregulation of SCL1A2 and -A3 on the two pages are a bit unclear - is "metabolic compensatory effects" meant to indicate uptake of aspartate and glutamate from dying cells?

Minor:

Given the large focus on SLC7A3, it is curious that its endogenous expression is not validated at the protein level. This would be a nice addition.

In Fig 3b (and the suppl fig 3c) it would be helpful with a scale indicating the meaning of the shades of purple (I suspect it has no meaning beyond distance from the center, but specification would avoid confusion).

P 9. Is SLC3A2 not upregulated at the mRNA level? If so, that would strengthen this statement.

LSA-2022-01404: Rebsamen, Girardi *et al.* Gain-of-function genetic screens in human cells identify SLC transporters overcoming environmental nutrient restrictions.

Response to Reviewer's comments:

We thank the reviewers for the careful assessment of our study and positive judgment. We are also grateful for the insightful comments, which we have fully addressed in the detailed point-by-point reply here below and in the revised manuscript.

Reviewer #1 (Comments to the Authors (Required)):

This manuscript describes an innovative gain of function screen to gain insight on the role of amino acid transporters. The authors show that SLC7A3 can overcome deficiencies in availability of both arginine and lysine in the environment. They also show that SLC7A5 can compensate for deficiencies of several neutral amino acids. Finally they show that SLC1A2/3 can compensate for deficiency of glutamine. This approach is useful to identify potential drug targets, for example in tumor cells, or to reveal the substrate specificity of transport proteins.

The experiments are well-controlled and clearly described. The paper is very well written. My main criticism relates to Figure 5. This experiment is described essentially as a confirmation of previous experiments. Yet, Fig 5b suggests that new insights on other compensatory pathways relating to arginine biogenesis may exist. Did you follow up on some of the other genes that were significantly up or down regulated in 5b? Could they be confirmed by dox induced overexpression as was done for SLC7A3? What about performing metabolic analysis for some?

We thank the Reviewer for the positive evaluation of our work and the comments. The results in Figure 5B show the enrichment at the single sgRNA level, confirming that multiple sgRNAs targeting the promoter of SLC7A3 provide a fitness advantage upon arginine limiting concentrations. When the results were aggregated at the gene level, as reported in Supplementary Table 9, only two genes resulted significantly enriched: SLC7A3 (3 sgRNAs, LFC 4) and DTNA, which encodes for Dystrobrevin alpha (2 sgRNAs, LFC 0.8). Given the much lower effect size observed with DTNA, we did not follow up on this hit further. Following the Reviewer's comment and the fact that no other gene reached statistical significance, we modified the Fig. 5B and removed the name of the other sgRNA previously indicated to avoid any risk of overinterpretation. While it is likely, as pointed out by the Reviewer, that other compensatory mechanisms may exist, these did not clearly manifest in our screen suggesting that SLC-mediated compensation is the prominent resistance mechanism, at least under these experimental conditions.

Minor: page 10, did you mean 70,290 sgRNAs?

Yes, we thank the Reviewer for pointing this out. We have now corrected this point in the revised manuscript.

Reviewer #2 (Comments to the Authors (Required)):

In this manuscript titled "Gain-of-function genetic screens in human cells identify SLC transporters overcoming environmental nutrient restrictions", Rebsamen et al. describe functional relationships between membrane transporters and amino acid limitations. They performed transcriptomics analyses and CRISPRa screens to identify transporter that can overcome specific amino acid restrictions. While approach and findings are of interest to amino acid metabolism field, some critical points remain to be addressed.

We thank the Reviewer for the positive assessment of our work. Replies to each of the raised points are found below.

Major points:

- While authors present qPCR data for mRNA level increases in SLC7A3 activation, membrane targeting and expression of SLC family membranes are also regulated at the protein level. Thus, activation of SLC7A3 expression by CRISPRa sgRNAs need to be shown at the protein level as well.

To address this important point, we have now identified an antibody (Sigma - Atlas Antibody: HPA003629) which detects endogenous SLC7A3 protein in immunoblot. After confirming its specificity using SLC7A3 cDNA-overexpressing cells (data now added as Fig S2E), we could now show increased SLC7A3 protein levels upon expression of gene-specific CRISPRa sgRNAs. The data have been added as panel 2d and indeed represent an improvement of the study.

- In Fig 2e, authors provide a competition assay for validation of SLC7A3's rescue effect on Arg restriction. They used CRISPRa sgRNAs for validation; however, these sgRNAs may activate downstream genes and have pleiotropic effects. That's why, cDNA overexpression of SLC7A3 should be included to show specific rescue effects in Arg restriction.

We now have included new CCA data in Figure 2G showing the effect of SLC7A3 cDNA expression upon arginine restriction. In line with the results obtained with SLC7A3-targeting CRISPRa sgRNAs, cDNA-mediated overexpression of SLC7A3 resulted in an increased fitness of the cells under Arg restriction. This confirms the gene-specific effect, which is also supported by the use of multiple independent sgRNAs in the CRISPRa assays.

- In Fig 2, authors perform GFP-based competition assays for validation. And, they nicely show that SLC7A3 confers competitive advantage. Does SLC7A3 overexpression also restore proliferative capacity of cells in Arg restriction back to complete media levels? How do SLC7A3 sgRNA cells grown in 1% Arg compare to those in reconstituted media.

As shown in Fig S2C, Arg restriction results in a strong selective pressure, which largely impairs cell proliferation. SLC7A3 overexpression, while conferring a competitive advantage, is not sufficient to restore fitness to a level comparable to culture in the complete media. Indeed, as shown in Fig 2F, the enrichment of SLC7A3 overexpressing cells in Arg restricted conditions is relatively slow and progresses with time. In line with this, we did not observe striking differences in short term proliferation assay (data not shown). This suggests that SLC7A3 overexpression preferentially affects cell survival rather than restoring proliferation.

- In the Results section, while describing Fig 3b, authors state "cytosolic aspartate concentrations determine cell survival upon glutamine starvation and that expression of SLC1A3 overcomes the effects of limiting glutamine availability". However, SLC1A3 can also transport Glutamate, that can

bypass the requirement of Gln and directly feed into TCA cycle. Authors need to consider this possibility as well.

We thank the Reviewer for this comment, which provides indeed a likely explanation. The text quoted refers to the findings of the mentioned references and not directly to our work. We have now included the potential explanation mentioned by the Reviewer in the discussion (page 13).

Minor points:

- P-values should be added to the following graphs: Fig 2e, 4a, 4b, 4d, 4e, 4f, 4g

We now added P-values in each of the mentioned panels. Details of the test used have been added to the corresponding figure legends.

- Can authors provide FACS data for color competition assays with a viability marker (DAPI, PI etc.) to show whether percentage of dead/live cell ratios over time in aa-restricted media?

We thank the Reviewer for this comment. We did not include viability markers in the color competition assay because, due to the assay set-up, this would not be particularly informative in our opinion. The color competition assays were performed over two weeks and included several (usually three) cell passaging steps, in which we removed dead cells and kept only adherent, live cells (i.e. discard media with dead cells, wash with PBS, trypsin treatment and re-seeding). The same procedure was performed before FACS analysis, and therefore resulted in further enrichment of live cells. Indeed, we performed a live gating step using FCS/SCC signals before monitoring GFP signals which resulted in over 90% of viable cells (data not shown). Therefore, a measurement of the dead/live ratio at the end of our color competition assay does not reflect the strong selection pressure induced by the aa-restricted media, which is shown in Fig S2C.

Reviewer #3 (Comments to the Authors (Required)):

General:

This work describes and validates the development of a genetic screen to identify SLC transporters, the upregulation of which is involved in supporting growth upon depletion of specific amino acids from the growth medium of HEK293 cells. Specifically, SLC7A3 is identified as upregulated under arginine and lysine depletion and able to overcome the growth-limitations induced by these conditions. Also SLC7A5 and SLC1A2 and -A3 were shown to support growth under nutrient-limiting conditions. The screen, which is based on CRISPR/Cas9-based transcriptional activation of a library of SLCs from their endogenous loci, is shown to clearly have the strength to identify novel roles of these key transporters, as long as their substrates are limiting for growth. Collectively, this is a very well-written manuscript describing a technically powerful study describing a method that will be potentially extremely useful to future studies of SLCs. My only regret is that the work raises numerous intriguing questions that are not addressed, such as the mechanisms of upregulation of each transporter as well as the normal physiological roles and interactions of the transporters. But the authors are well aware of this and do not claim otherwise - the manuscript is mainly intended to describe this elegant tool for the community. Thus, I have by far mostly praise for the work, but some queries and suggestions are provided below.

We thank the Reviewer for the very positive evaluation of our study and to appropriately assess the scope of the current manuscript.

Specific:

p. 3. As mentioned in the introduction, autophagy is powerfully regulated by starvation, but in the rest of the manuscript, this is not further mentioned, and the transporters identified are mainly plasma membrane localized. It would be interesting to describe to what extent a role for SLCs localized to autophagosomes/lysosomes was observed? Perhaps it would be worth providing an overview, graphically or otherwise, of which organelles, the identified transporters localized to?

We thank the Reviewer for his/her comments. Most of the transporters identified are localized to the plasma membrane. As suggested by the Reviewer, we have now included an overview (Fig S3E) of the reported localization (according to Meixner et al, 2020 and the COMPARTMENTS database) of the SLCs enriched in our screens.

p. 5. "To identify the transcriptional programs induced by nutrient limitation..." is perhaps a bit exaggerated, since the authors actually do not go beyond the TF analysis in Fig 1c, and the enrichment of ATF4 was fully expected from the literature. Hence, I suggest to rephrase. It would in my view also be interesting to comment on the apparently unique TF enrichment profile after Ile starvation (with much less response for instance after starvation for all AAs, which is offhand surprising, at least to me) and perhaps on the strong C/EBP enrichment after Cys starvation.

We agree with the Reviewer's comment. As suggested, we have now rephrased the relevant sentence and added an additional sentence commenting on the Cys CEBPB enrichment, which is consistent with previous reports in another cell line (Lee et al, Physiol Genomics, 2008). In the case of Ile, the enrichments observed in HEK293 cells for additional TF beyond ATF4 were much less significant than ATF4 and the TF enrichment profile in HeLa cells did not appear to be significantly different for Ile than in the other conditions (Fig S1C). For these reasons, in the manuscript we would prefer not to speculate further on this point.

Fig. 1d:

1) Isn't it surprising (or am I just missing something) that SLC6A14 and -15, which are so strongly enriched in this condition in the CRISPR screen, are not among the transporters upregulated when all amino acids are depleted in the mRNA sequencing analysis? Please discuss.

This is an interesting observation and we are grateful to the Reviewer for pointing this out. In our transcriptomics datasets, we did not detect SCL6A14, possibly pointing to a cell-type specific regulation. SLC6A15 transcripts were however detected in these cell lines, suggesting that either the different experimental setups used (16h nutrient starvation for the transcriptomics analysis vs week-long genetic screen in depleted nutrient conditions) or the increased efficiency of the CRISPRa approach in overriding possible regulatory mechanisms were responsible for this discrepancy. Interestingly, this further highlights the potential of our g-o-f screening approach to identify nutrient-SLC interactions not readily accessible to transcriptomics studies.

2) I know that this is not the goal of this work, but it would perhaps be worth also commenting on the many strongly DOWN-regulated SLCs, which also seem to be shared among many of the starvation conditions - do they make physiological sense, given what is known about them in the literature? And

on another, to me surprising note: why is the methionine transporter SLC43A2 so strongly downregulated uniquely when all AAs are depleted?

As the Reviewer pointed out, several SLCs are also downregulated upon nutrient starvation. This is an interesting point that warrants further investigation by, for example, systematic I.o.f. approaches, even if these could be challenging given the redundancy intrinsic to the system. It is also likely that upregulation and downregulation of a given set of transporters aims at expressing the optimized set of SLCs (in terms of e.g. affinity/capacity properties, counterions involved) for the given set of substrates. Regarding SLC43A2, this is indeed interesting, but to speculate further we believe that only a detailed functional investigation would allow us to bring forward a meaningful hypothesis, which, as mentioned by the referee, is beyond the scope of the current study. It may help to point out, that the bioenergetic costs of making membrane proteins, especially, glycosylated ones, is particularly high. Thus, upon nutrient shortage, downregulation of SLC genes may be a cost-saving measure.

P9/Fig. 3b: As stated by the authors the mitochondrial ornithine and citrulline transporter SLC25A15 is, interestingly, upregulated in many of the conditions tested - but I do not understand the argument for why that is so ("as a result of sgRNA depletion in the fully reconstituted media condition" - please rephrase - and if you can speculate on the physiological logic of this upregulation that would be very nice.

We agree with the referee regarding the fact that this statement was not clear. We have now added the profile of the SLC25A15 sgRNA across the different conditions screened in new Fig S3C. These data show that the abundance of two SLC25A15 sgRNAs (sg1 and sg4) in the fully reconstituted media is much lower than what is observed in any of the other condition tested, including at time 0 and in full media (the two other control conditions). As sgRNA enrichment is calculated by comparing nutrient-deprived and fully reconstituted media conditions, all nutrient-deprived conditions will automatically show enrichment for SLC25A15 even though the sgRNAs themselves are not enriched in the nutrient-deprived conditions compared to other, non-scoring, SLC sgRNAs. We have now edited the corresponding sentence to clarify this point. Because of this effect, we suspect that the enrichment of SLC25A15 in most conditions is likely the consequence of these technical reasons, rather than a true biological response.

Fig 3c-d vs Fig 4a:

When only sg3, -4, and -5 are enriched (Fig 3c-d), possibly (as suggested by the authors) because the other sgs were less efficient in triggering transcriptional activation, why are sg1-, -2, and -3 used in the color competition assay, and how does the above interpretation fit with the fact that they do increase cell fitness?

We thank the Reviewer for raising this point and apologize for the mislabelling. The sgRNA annotation between the two figures was not consistent and it has now been corrected. The SLC7A3 sgRNAs annotated as sg1, 2, 3 in the original Figure 4A correspond to the sg3, 4 and 5 of the SLC-SAM library. The numbering of SLC7A3 sgRNAs across the revised manuscript is now reflecting the usage of the sg3, 4 and 5 from the library.

P 10 vs p 13. The explanations offered for the upregulation of SCL1A2 and -A3 on the two pages are a bit unclear - is "metabolic compensatory effects" meant to indicate uptake of aspartate and glutamate from dying cells?

This is indeed in our opinion the most likely hypothesis, as we mentioned in the discussion. Given that we cannot formally exclude other possible indirect mechanisms, we opted to use the admittedly generic formulation of "metabolic compensation". As mentioned, future detailed investigations to

clarify the mechanism(s) responsible for the observed phenotype are of interest, but are not straightforward and go beyond the scope of this manuscript.

Minor:

Given the large focus on SLC7A3, it is curious that its endogenous expression is not validated at the protein level. This would be a nice addition.

The Reviewer is absolutely right. As mentioned also in the reply to Reviewer 1, to address this point we have now identified an antibody (Sigma - Atlas Antibody: HPA003629) which detects endogenous SLC7A3 protein in immunoblot. After confirming specificity using SLC7A3 cDNA-overexpressing cells (new Fig S2E), we could now show also increased SLC7A3 protein levels upon expression of gene-specific CRISPRa sgRNAs (new Fig 2D).

In Fig 3b (and the suppl fig 3c) it would be helpful with a scale indicating the meaning of the shades of purple (I suspect it has no meaning beyond distance from the center, but specification would avoid confusion).

Indeed, the color shade reflects the distance from the center of the plot and therefore the degree of significance of the effect observed. We have now modified the figure legends to clarify this.

P 9. Is SLC3A2 not upregulated at the mRNA level? If so, that would strengthen this statement.

We thank the Reviewer for this suggestion. We have now added qPCR analysis of SLC3A2 transcripts in sgSLC7A5-expressing cells which show that overall SLC3A2 mRNA levels are not strongly upregulated (new Fig S4A), supporting therefore our hypothesis.

August 11, 2022

RE: Life Science Alliance Manuscript #LSA-2022-01404R

Prof. Giulio Superti-Furga
CeMM Research Center for Molecular Medicine
CeMM Research Center for Molecular Medicine of the Austrian Academy of Sciences
Lazarettgasse 14
Vienna 1090
Austria

Dear Dr. Superti-Furga,

Thank you for submitting your revised manuscript entitled "Gain-of-function genetic screens identify SLC transporters overcoming nutrient restrictions". We would be happy to publish your paper in Life Science Alliance pending final revisions necessary to meet our formatting guidelines.

- please consult our manuscript preparation guidelines <https://www.life-science-alliance.org/manuscript-prep> and make sure your manuscript sections are in the correct order
- please use the [10 author names, et al.] format in your references (i.e. limit the author names to the first 10)

Figure Check:

- S4 G, H need scale bars

A. FINAL FILES:

B. MANUSCRIPT ORGANIZATION AND FORMATTING:

Sincerely,

Reviewer #1 (Comments to the Authors (Required)):

The authors have addressed my prior concerns appropriately and no further issues remain. This paper describes an innovative gain of function screen that provides new insights on functional interactions between amino acid transporters. Furthermore, the methods used for screening transcriptionally activated genes is widely applicable. The library activating 388 SLC genes is an incredible community resource. Overall, this is a highly impactful contribution.

Reviewer #2 (Comments to the Authors (Required)):

Authors have significantly improved the manuscript by including necessary control experiments and revising the text. All points raised during the review process have been satisfactorily answered. Particularly, they identified an antibody for SLC7A3 and verified that CRISPRa guides can upregulate its protein levels by performing immunoblots. Additionally, authors now included a SLC7A3 cDNA overexpression experiment to show that it is sufficient to overcome Arg restriction, demonstrating its specificity regardless of gene activation method, CRISPRa or cDNA overexpression. They amended the main text to integrate these results; and added an interpretation of SLC1A3-mediated Glu transport as a possible explanation for rescue under Gln deprivation.

Reviewer #3 (Comments to the Authors (Required)):

I am very pleased with the revisions and responses made by the authors, which have significantly improved the manuscript. I have no further comments.

August 25, 2022

RE: Life Science Alliance Manuscript #LSA-2022-01404RR

Prof. Giulio Superti-Furga
CeMM Research Center for Molecular Medicine of the Austrian Academy of Sciences
Lazarettgasse 14
Vienna 1090
Austria

Dear Dr. Superti-Furga,

Thank you for submitting your Research Article entitled "Gain-of-function genetic screens identify SLC transporters overcoming nutrient restrictions". It is a pleasure to let you know that your manuscript is now accepted for publication in Life Science Alliance. Congratulations on this interesting work.

DISTRIBUTION OF MATERIALS:

Again, congratulations on a very nice paper. I hope you found the review process to be constructive and are pleased with how the manuscript was handled editorially. We look forward to future exciting submissions from your lab.

Sincerely,
